# Graphene Oxide as a Sensing Material for Gas Detection Based on Nanomechanical Sensors in the Static Mode

**Gaku Imamura [1,*]**, **Kosuke Minami [2]**, **Kota Shiba [3]**, **Kissan Mistry [4,5]**,
**Kevin P. Musselman [4,5]**, **Mustafa Yavuz [4,5]**, **Genki Yoshikawa [3,6]**,
**Koichiro Saiki [7]** and **Seiji Obata [8]**

[1] World Premier International Research Center Initiative (WPI), International Center for Materials Nanoarchitectonics (MANA), National Institute for Materials Science (NIMS), 1-1 Namiki, Tsukuba, Ibaraki 305-0044, Japan

[2] International Center for Young Scientists (ICYS), National Institute for Materials Science (NIMS), 1-1 Namiki, Tsukuba, Ibaraki 305-0044, Japan; MINAMI.Kosuke@nims.go.jp

[3] Research Center for Functional Materials, National Institute for Materials Science (NIMS), 1-1 Namiki, Tsukuba, Ibaraki 305-0044, Japan; SHIBA.Kota@nims.go.jp (K.S.); YOSHIKAWA.Genki@nims.go.jp (G.Y.)

[4] Department of Mechanical and Mechatronics Engineering, University of Waterloo, 200 University Ave. West, Waterloo, ON N2L 3G1, Canada; kissan.mistry@uwaterloo.ca (K.M.); kevin.musselman@uwaterloo.ca (K.P.M.); myavuz@uwaterloo.ca (M.Y.)

[5] Waterloo Institute for Nanotechnology, 200 University Ave. West, Waterloo, ON N2L 3G1, Canada

[6] Materials Science and Engineering, Graduate School of Pure and Applied Science, University of Tsukuba, 1-1-1 Tennodai, Tsukuba, Ibaraki 305-8571, Japan

[7] Graduate School of Frontier Sciences, University of Tokyo, 5-1-5 Kashiwanoha, Kashiwa, Chiba 277-8561, Japan; saiki@edu.k.u-tokyo.ac.jp

[8] Research Core for Interdisciplinary Sciences, Okayama University, 3-1-1 Tsushimanaka, Kita-ku, Okayama, Okayama 700-8530, Japan; obata.s@okayama-u.ac.jp

\* Correspondence: IMAMURA.Gaku@nims.go.jp; Tel.: +81-(0)-29-860-4988

**Abstract:** Graphene is a key material for gas sensing applications owing to its high specific surface area and vast chemical modification potential. To fully utilize the potential of graphene, a sensing platform independent of conductive properties is required. In this study, we employed membrane-type surface stress sensors (MSS)—A kind of nanomechanical sensor operated in the static mode—As a sensing platform and utilized graphene oxide (GO) as a gas sensing material. MSS detect surface stress caused by gas sorption; therefore, chemically modified graphene with low conductivity can be utilized as a gas sensing material. We evaluated the sensing performance of a GO-coated MSS by measuring its responses to five gases. We demonstrated with the GO-coated MSS the feasibility of GO as a gas sensing material for static mode nanomechanical sensors and revealed its high selectivity to water vapor. Moreover, we investigated the sensing mechanism of the GO-coated MSS by comparing it with the sensing performance of MSS coated with reduced graphene oxide and graphite powder and deduced key factors for sensitivity and selectivity. Considering the high sensitivity of the GO-coated MSS and the compact measurement system that MSS can realize, the present study provides a new perspective on the sensing applications of graphene.

**Keywords:** graphene oxide; gas detection; nanomechanical sensors; Membrane-type Surface Stress Sensors (MSS)

## 1. Introduction

Since being first isolated in 2004, graphene has been widely studied in many fields, ranging from fundamental science to industrial applications [1,2]. Owing to its high specific surface area and its potential for chemical functionalization, graphene is expected to be a key material for sensing applications. Many studies have reported its potential as a sensing material for chemical species such as gases, ions, and proteins [3–6]. Most of these studies have described sensors based on detection of resistance changes associated with sorption of analytes, namely, chemiresistive sensors and field-effect transistor sensors. In these types of sensors, resistance changes in graphene are directly detected as sensing signals. The sensing materials in such sensors, however, are required to have high conductivity or high carrier mobility; otherwise, sensing signals cannot be obtained from sorption of analytes. Although pristine graphene is highly conductive and has a high electron mobility, chemical modification, including functionalization and doping, deteriorates its conductive properties. Owing to this trade-off relationship, the advantages of graphene cannot be fully utilized in such sensing applications.

To resolve this issue, a sensing platform that does not depend on conductive properties is required. One such sensing platform is nanomechanical sensors in the static mode. Static mode nanomechanical sensors detect changes in mechanical properties, such as deflection and stress, caused by sorption of analytes [7]. The most typical structure of a static mode nanomechanical sensor is a microcantilever coated with a sensing material. When the sensor is exposed to analytes, the sensing material absorbs the analytes and mechanically swells, resulting in the deflection of the cantilever and the generation of surface stress. Such deflection or surface stress is transduced to measurable sensing signals. Thus, any solid material can be utilized as a sensing material regardless of its conductive properties. In 2011, Yoshikawa et al. reported a new type of static mode nanomechanical sensor—The membrane-type surface stress sensor (MSS) [8]. Because of its unique structure, MSS can efficiently detect surface stresses associated with the swelling/shrinking of a sensing material by measuring resistance changes of embedded piezoresistors, which is sensitive to mechanical stress (Figure 1). Owing to the novel working principle, the MSS achieved both high sensitivity and compactness. In addition, the MSS exhibits much higher robustness in the topography of a sensing material, leading to higher repeatability than a cantilever-type sensor [9,10].

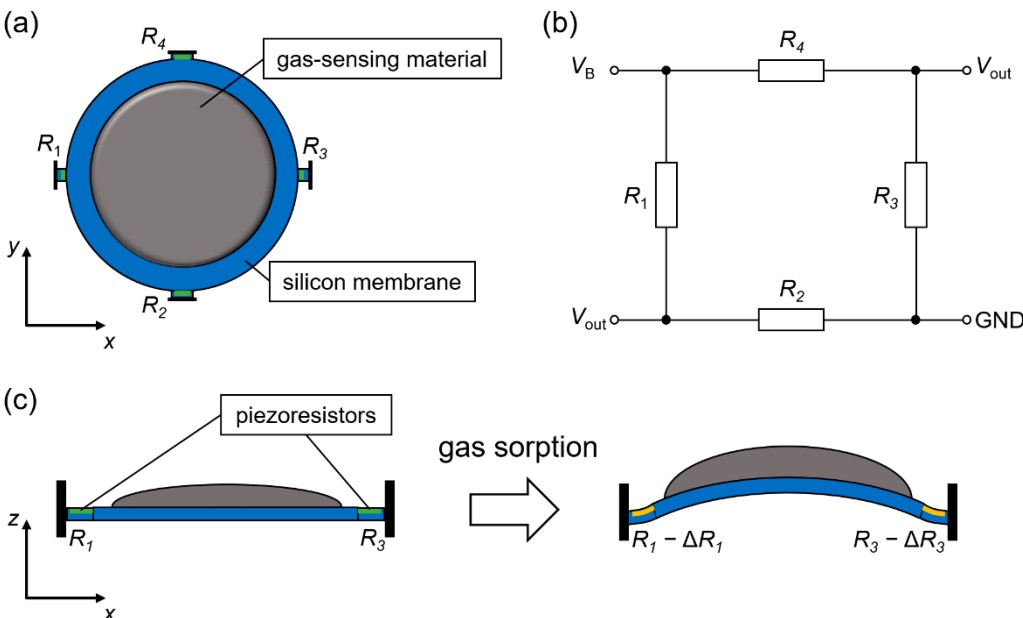

**Figure 1.** Structure of the membrane-type surface stress sensor (MSS). (**a**) Top view of the MSS. (**b**) Circuit diagram of the MSS. (**c**) Side view of the MSS.

One of the greatest merits of using graphene and its derivatives as a sensing material in static mode nanomechanical sensors is its high Young's modulus. According to the analytical model proposed by Yoshikawa, sensor sensitivity strongly depends on the Young's modulus of the sensing materials; a high Young's modulus is usually preferable when isotropic internal strain in sensing materials is constant [11]. A theoretical model formulated by Wenzel et al. also suggests that a sensing material with a high Young's modulus yields high sensitivity [12–14]. As the Young's moduli of graphene and its derivatives are remarkably high [15–17], high sensitivity is expected for graphene-based nanomechanical sensors in the static mode. Even though there are other types of sensors that do not depend on conductive properties of a sensing material (e.g., quartz crystal microbalance, surface acoustic wave, and nanomechanical sensors in the dynamic mode), such sensors do not utilize the high Young's moduli of graphene and its derivatives [18–21].

Thus, static mode nanomechanical sensors can fully utilize the advantages of graphene and its derivatives as a sensing material. However, the literature reporting utilization of graphene as a sensing material in static mode nanomechanical sensors is limited; to the best of our knowledge, only two studies have demonstrated utilization of graphene in such a way. Basu et al. reported a functionalized reduced graphene oxide (rGO)-multiwalled carbon nanotube nanocomposite material for the detection of cholesterol by using a cantilever-type sensor [22]. Yao et al. reported fabrication of a stress-based humidity sensor based on a graphene oxide (GO)-silicon (Si) bilayer structure [23]. Owing to the limited number of previous studies, the sensing mechanism of graphene-based nanomechanical sensors in the static mode has not been well clarified, giving rise to a lack of guidelines for optimizing and modifying sensing properties. Thus, exploring the working principles behind graphene-based nanomechanical sensors in the static mode can expand perspectives on sensing applications that utilize graphene and its derivatives.

In this study, we focused on GO as an example of a graphene derivative. GO is a chemically modified graphene derivative produced by chemically oxidizing graphene or graphite and has oxygen-containing functional groups. As GO can be synthesized through a simple solution process, it is expected to be a key material in the practical application of graphene [24]. The objectives of this study were to develop a nanomechanical sensor system for gas detection using membrane-type surface stress sensors (MSS) as a static mode nanomechanical sensing platform, because of their high sensitivity and compactness [8,25], and to investigate the sensing mechanism of GO as a sensing material.

## 2. Materials and Methods

### 2.1. Materials

A GO suspension synthesized by the Hummers method was purchased from Graphene Supermarket (New York, NY, USA). The GO suspension was diluted with ultrapure water (MilliQ) to a concentration of 0.29 g/L, and then sonicated for 2 h. All the rGO samples used in this study were synthesized by annealing a GO-coated MSS or a GO-coated Si substrate at 200 °C in vacuum for 2 h. Graphite powder was provided by Nippon Graphite Industries (Otsu city, Japan). The graphite powder was dispersed in 1,1,2,2-tetrachloroethane (TCE) and sonicated for 2 h, resulting in a stable dispersion of graphite powder. The concentration of the graphite powder in TCE was 0.29 g/L. Organic solvents for gas sensing measurements (i.e., acetone, ethanol, heptane, and ethyl acetate) were purchased from Sigma-Aldrich (St. Louis, MO, USA), Tokyo Chemical Industry (Tokyo, Japan), and Wako Pure Chemical Industries (Osaka, Japan), and used without further purification. The standard gases used in this study were provided by Sumitomo Seika Chemicals Company (Tokyo and Osaka, Japan).

## 2.2. Materials Characterization

### 2.2.1. Optical Microscopy (OM)

An upright microscope (Eclipse Ni, Nikon Corporation, Tokyo, Japan) was used for OM observation. Images were recorded with a microscope camera (DS-Ri2, Nikon Corporation, Tokyo, Japan). Samples were observed in an ambient condition.

### 2.2.2. Scanning Electron Microscopy (SEM)

Both GO and graphite powder were observed with S-4800 (Hitachi, Ltd., Tokyo, Japan). The GO solution and graphite dispersion were manually drop-cast onto Si substrates and well dried in air before observation.

### 2.2.3. Transmission Electron Microscopy (TEM)

GO was observed with JEM2100F (JEOL Ltd., Tokyo, Japan). A diluted GO solution was drop-cast onto a TEM grid (Micro Grid NP-C15 (based on Cu 150P, Lacy Carbon Film, Okenshoji Co., Ltd., Tokyo, Japan) and dried in vacuum for more than 10 h before observation.

### 2.2.4. Infrared (IR) Spectroscopy

IR measurements were performed with an FT-IR microscope (IR NICOLETiS5, Thermo Fisher Scientific, Waltham, MA, USA). GO and graphite samples were prepared by manually drop-casting each solution or dispersion onto a Si substrate. A GO-coated Si substrate was annealed at 200 °C in vacuum for 2 h to form an rGO sample. Spectra were obtained in the reflection mode. The measurement system was purged with a nitrogen flow before and during measurements.

### 2.2.5. Raman Spectroscopy

Raman spectra were obtained with NRS3100 (JASCO Corporation, Tokyo, Japan). The excitation wavelength was 532 nm. Measurements were performed on GO, rGO, and graphite powder on Si substrates. GO and graphite powder were delivered onto Si substrates by inkjet-spotting. An rGO sample was prepared by annealing a Si substrate, on which GO was delivered by inkjet-spotting at 200 °C in vacuum for 2 h.

### 2.2.6. Water Contact Angle Measurements

Water contact angle measurements were performed with a contact angle meter (P200A, Meiwafosis Co., Ltd., Tokyo, Japan). Silicon substrates coated with GO and graphite powder were prepared by coating Si substrates with the GO solution and graphite dispersion by manual drop-casting. A GO-coated Si substrate was annealed at 200 °C in a vacuum for 2 h to form an rGO-coated substrate.

### 2.2.7. Thermogravimetric Analysis (TGA)

The GO, rGO, and graphite powder used in this study were analyzed by thermogravimetric analysis (TGA) (STA 2500 Regulus, NETZSCH, Selb, Germany). A GO sample was prepared by drying a GO suspension. An rGO sample was prepared by drying a GO suspension and annealing it at 200 °C in a vacuum for 2 h. A graphite powder sample was obtained by drying a graphite powder dispersion. Samples were heated at a rate of 10 °C/min in dry air.

### 2.2.8. Dynamic Light Scattering (DLS)

Dynamic light scattering (DLS) measurements were performed with Zetasizer Nano ZSP (Malvern Panalytical Ltd., Malvern, UK) equipped with a He–Ne laser operating at 4 mW power and 633 nm wavelength and a computer-controlled correlator at a 173° accumulation angle. Measurements were

performed only on GO; the sizes of the graphite flakes in the graphite powder were too large for DLS measurements, and stable dispersion of rGO could not be obtained.

### 2.3. MSS

MSS is an MEMS-based sensor made of Si. The fabrication process of MSS has been shown previously [26]. MSS has a distinctive structure; a thin membrane is suspended by four beams in which piezoresistors ($R_1$, $R_2$, $R_3$, and $R_4$) are embedded (Figure 1a). The piezoresistors compose a full Wheatstone bridge (Figure 1b). The membrane is coated with a gas sensing material. Surface stress caused by swelling or shrinking of the gas sensing material is detected as resistance changes by the piezoresistors (Figure 1c) [8,25]. The signal output of MSS ($V_{out}$) is described as follows,

$$V_{out} = \frac{V_B}{4}\left(\frac{\Delta R_1}{R_1} - \frac{\Delta R_2}{R_2} + \frac{\Delta R_3}{R_3} - \frac{\Delta R_4}{R_4}\right) \tag{1}$$

where $V_B$ is the bridge voltage applied to the Wheatstone bridge circuit. In this study, $V_B = -0.5$ V was applied with a digital-to-analog converter module (NI-9269, NI, Austin, TX, USA). The signal outputs were measured with an analog-to-digital converter (NI-9214, NI, Austin, TX, USA). The sampling frequency was set at 20 Hz. MSS chips were supplied by NanoWorld AG (Neuchâtel, Switzerland).

An inkjet spotter (LaboJet-500SP, MICROJET Corporation, Shiojiri, Japan) was used to coat the membrane with sensing materials. Details of the coating method are presented in our previous studies [14,27–30]. The prepared GO suspension and graphite powder dispersion were used for inkjet spotting. The total number of shots was fixed at 300, while the volume of one droplet was approximately 300 pL. MSS coated with rGO was prepared by annealing the GO-coated MSS in a vacuum at 200 °C for 2 h.

### 2.4. Surface Profile

The surface topography of the materials coating the Si substrates was measured by a contact surface profiler (DektakXT, Bruker, Billerica, MA, USA). Inkjet spotting was used to coat the Si substrate with the GO suspension according to the same procedure used to prepare GO-coated MSS. To investigate changes in thickness due to the vacuum annealing process, 1 µL of GO suspension was cast on a Si substrate 10 times, and then the GO-coated substrate was annealed at 200 °C for 2 h in a vacuum.

### 2.5. Gas Sensing Measurements

2.5.1. Measurements of Solvent Vapors

Gas sensing measurements were performed with a gas flow line equipped with two mass flow controllers (MFCs; SEC-N112, HORIBA STEC, Co., Ltd., Kyoto, Japan). To provide a sample solvent vapor to an MSS chip, one MFC (MFC1) was connected to a screw vial containing a sample solvent for nitrogen gas from MFC1 to pass through (Figure 2a). In this study, water, acetone, ethanol, heptane, and ethyl acetate were used as sample solvent vapors by considering their chemical characteristics. The nitrogen, which was almost saturated with the solvent vapor, was then diluted with pure nitrogen from the other MFC (MFC2). The diluted vapor was carried to a chamber containing an MSS chip. The MFCs, screw vial, and chamber were connected with Teflon tubes. The measurement temperature was kept at 25 °C. The total gas flow rate of the two mass flow controllers was set at 100 standard cubic centimeters per minute (SCCM). First, only nitrogen was supplied to the chamber (MFC1:MFC2 = 0:100) for 30 s. Then, the sample gas (MFC1:MFC2 = 20:80) was injected into the chamber for 30 s, followed by a pure nitrogen purge (MFC1:MFC2 = 0:100) for 30 s. The concentration of the sample gases can be calculated from the vapor pressures of the solvents: 6000, 61,000, 16,000, 12,000, and 25,000 ppm for water, acetone, ethanol, heptane, and ethyl acetate, respectively [31]. This cycle of sample gas injection and pure nitrogen purge was repeated four times. In this cycle, sensing signals in the first

cycle tend to be disturbed by residual gases from the previous measurements. To avoid such effects and enhance repeatability of the experiments, the MSS chip was purged with water vapor, which was confirmed to efficiently promote the desorption of most gaseous molecules because of the smaller size and the higher affinity of a water molecule to the receptor materials than most other molecules, as well as a pure nitrogen, before every measurement. Accordingly, the sensing signals in the first cycle tend to also be disturbed by these water molecules in addition to the residual gases from the previous measurements. As the sensing signals are stabilized for the latter cycles, the last three cycles were used for analysis. Baseline offset values—the signal output at the second sample gas injection—were subtracted from signals.

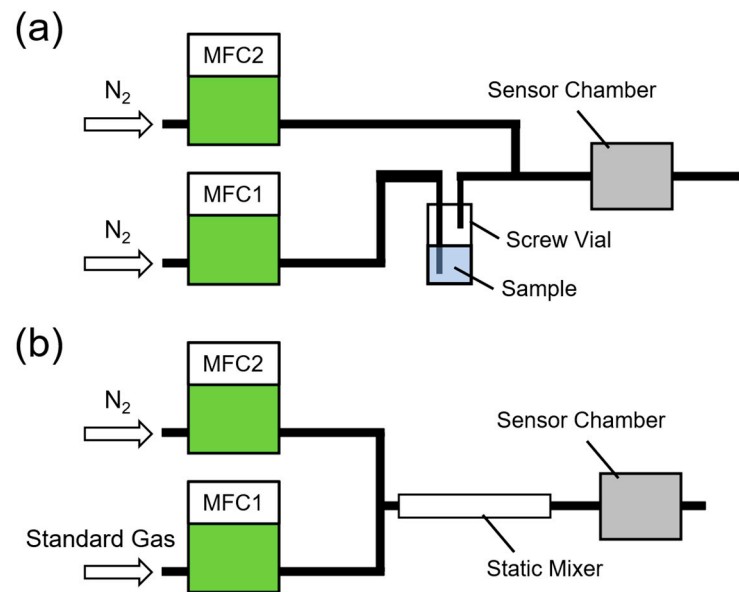

**Figure 2.** Gas sensing measurement systems. Schematic illustrations of the measurement systems for (**a**) solvent vapors and (**b**) standard gases.

### 2.5.2. Measurements of Low Concentration Gases

To investigate the sensor responses for low concentration gases and evaluate the limit of detection (LoD), gas sensing measurements with standard gases were performed with a separate measurement system. A similar configuration was employed for the standard gas measurements (Figure 2b). Standard gas and pure nitrogen cylinders were connected to MFC1 and MFC2, respectively, and the gases were supplied to the chamber. The total gas flow rate was set at 30 SCCM. The cycle of standard gas injection (MFC1:MFC2 = 30:0) and nitrogen purge (MFC1:MFC2 = 0:30) was repeated five times after the initial nitrogen purge (MFC1:MFC2 = 0:30) for five minutes. The time interval of the standard gas injection and nitrogen purge was set at five minutes. To avoid gas adsorption inside the gas flowline, all components including pipes were made of stainless steel and kept at 37 °C. In both gas measurement systems, control of gas flow rates and electrical readout of sensing signals were accomplished with a custom program based on LabVIEW (NI, Austin, TX, USA).

## 3. Results

### 3.1. GO-, rGO-, and Gr-Coated MSS

Figure 3a shows an OM image of GO cast on a silicon substrate. GO can be seen as the blurry blown objects. Although a few large flakes are observed, most are too small to be seen with the optical microscope. The SEM image as shown in Figure 3b indicates that GO flakes were aggregated. The sheet-like structure of a GO flake was confirmed with the TEM images; a thin sheet of GO can be seen in Figure 3c, whereas the wrinkles of the sheet were observed in Figure 3d. To evaluate the size distribution of the GO flakes, DLS measurements were performed on the GO suspension. The results indicate that a flake size of less than 1 µm was dominant in the GO suspension (Figure 3e). See "Interpretation of DLS Result" in the Supplementary Materials for the interpretation of the DLS measurements. Figure 4a shows an OM image of the MSS coated with GO. From the results of the surface profiler, it was revealed that the thickness of the coating is not uniform; the thickness varies in the range between 0 to 1 µm (details are shown in "Surface Profiles of GO and rGO Coatings on Si Substrates" in the Supplementary Materials). Although the coating on the MSS membrane is not uniform, it was already demonstrated that roughness or coverage of the coating does not significantly affect the sensing signals as long as the total volume of the coating material is constant [9,10]. Figure 4c,d shows the IR and Raman spectra (black lines), respectively, of the GO. Several peaks that are characteristic of GO can be seen in Figure 4c; broad, strong peaks from the O–H stretching mode, C=O stretching mode, and C–O–C stretching mode can be observed at 3300, 1700 and 1080 cm$^{-1}$, respectively [32]. Thus, the GO contained oxygen-containing functional groups, such as hydroxy, epoxide, and carbonyl groups. The Raman spectrum of the GO showed features typical of sp$^2$-hybridized carbon networks containing a large number of defects: the G-band (1580 cm$^{-1}$) and the D-band (1350 cm$^{-1}$) were widened, and their intensities were almost the same (Figure 4d) [33,34]. In addition, the 2D-band and the G+D band weakly appeared at approximately 2700 and 2900 cm$^{-1}$, respectively.

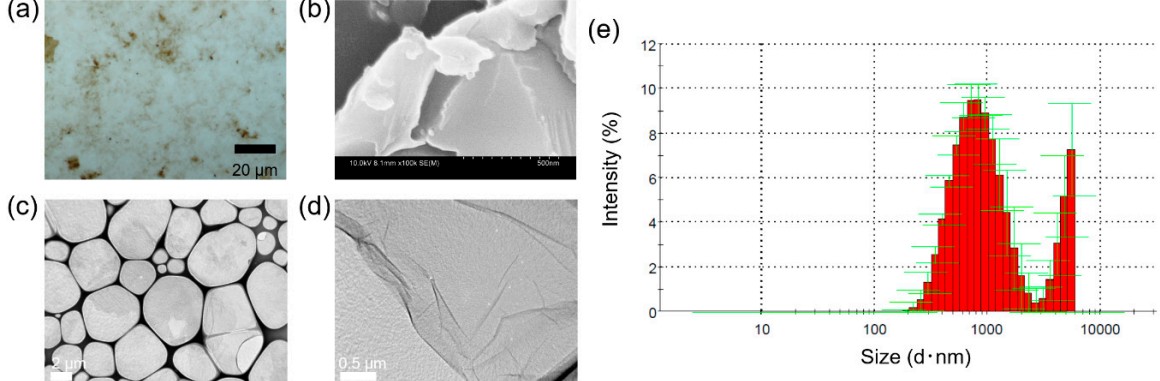

**Figure 3.** Properties of GO. (**a**) Optical microscope image of GO coating on a Si substrate. (**b**) SEM image of GO. (**c**,**d**) TEM image of GO. (**e**) The intensity distribution of the GO suspension obtained by dynamic light scattering (DLS) measurement.

Gas sensing measurements were performed with the gas flow line shown in Figure 2a. Figure 5a shows the results of the measurements. The GO-coated MSS showed clear sensing responses to the vapors, showing that GO can be used as a gas sensing material for nanomechanical sensors in the static mode. Another feature of the results is its selectivity—the GO-coated MSS responded selectively to water. We also confirmed by OM that the GO-coating was not changed after the measurements.

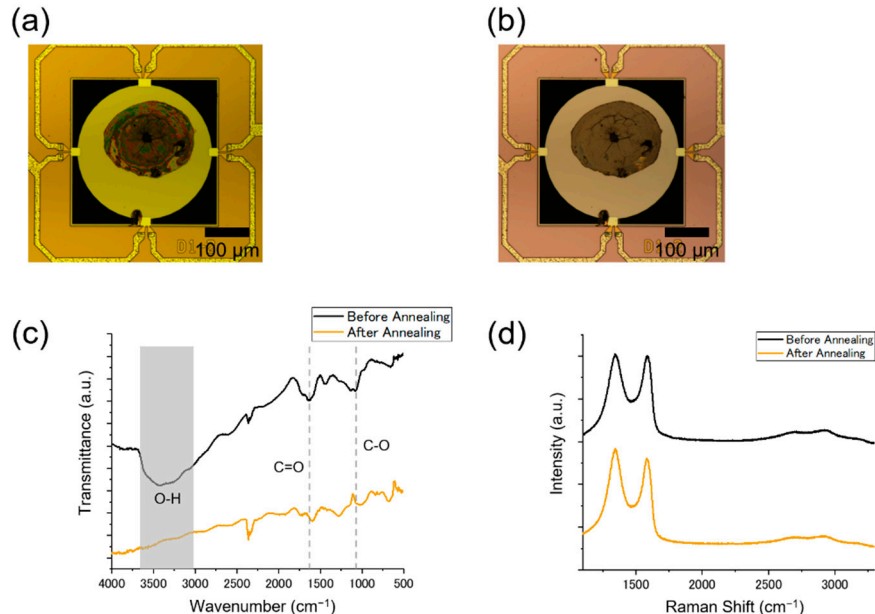

**Figure 4.** Effect of vacuum annealing on GO. Optical microscope images of MSS coated with GO (**a**) before and (**b**) after vacuum annealing. (**c**) IR spectra of GO and rGO coatings on MSS. The split peak at 2340 cm$^{-1}$ originates from $CO_2$. (**d**) Raman spectra of GO and rGO coatings on MSS.

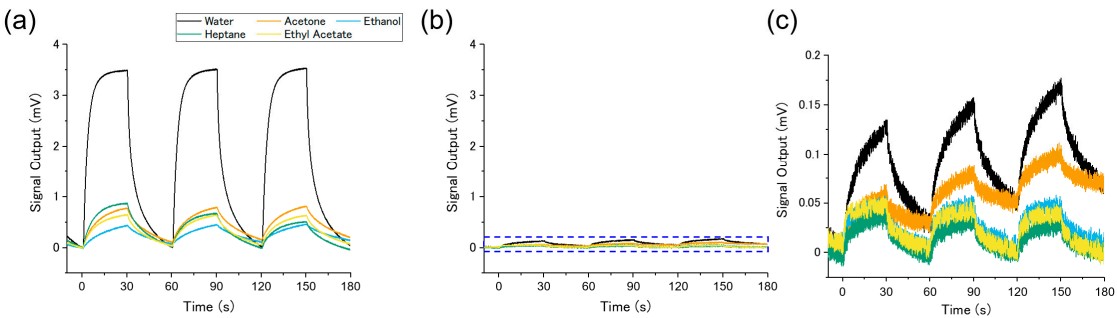

**Figure 5.** Results of gas sensing measurements for GO- and rGO-coated MSS. Sensing signals of (**a**) GO-coated MSS and (**b**) rGO-coated MSS for the solvent vapors. (**c**) Enlarged figure of the sensing signals of rGO-coated MSS indicated by the dashed rectangle in (**b**).

This superior sensing performance of the GO-coated MSS was changed by reduction of GO. To explore the sensing mechanism of the GO-coated MSS, we also investigated the sensing properties of rGO-coated MSS. TGA provided quantitative information for evaluating the thermal stability of GO. Figure 6 shows the TGA curve of the GO sample (black line). Weight losses below 100 °C and near 200 °C are attributed to losses of water retained in GO and removal of oxygen-containing functional groups, respectively (Figure 6a) [35–37]. Thus, the annealing temperature for reducing GO was set at 200 °C. Figure 4b shows an optical microscope image of the GO-coated MSS after vacuum annealing. Compared with that of the GO-coated MSS, the contour of the coating was unchanged; the coating did not crumble or peel from the substrate membrane during the vacuum annealing process. However, surface profiler measurements revealed that the thickness of the GO coating decreased after vacuum annealing. To investigate the effects of vacuum annealing on the thickness of the GO coating, we measured the surface profiles of GO coating on a Si substrate before and after vacuum annealing. After vacuum annealing, the thickness of the coating decreased by ~25%, as shown in "Surface Profiles of GO and rGO coatings on Si Substrates" in the Supplementary Materials (Figure S2).

Thus, the thickness of GO coating MSS should have also been decreased by vacuum annealing. The IR spectrum of GO after vacuum annealing is shown in Figure 4c (orange line). Compared to those in the IR spectrum taken before vacuum annealing, peaks related to oxygen-containing functional groups (3300, 1700, and 1080 cm$^{-1}$) weakened or disappeared in the IR spectrum taken after vacuum annealing. The disappearance of the peak assigned to the O–H stretching mode (3400 cm$^{-1}$) indicates that the hydroxy groups and carboxy groups in GO were reduced by vacuum annealing. The peaks related to C=O and C–O–C can still be seen in the IR spectrum after vacuum annealing, though both were weaker. The result of TGA for rGO is shown in Figure 6b (orange line). Although weight loss of rGO is much smaller than that of GO, rGO also shows weight loss near 200 °C, with a total weight loss of 0.8%. This result supports the IR results wherein the residual oxygen-containing functional groups still existed in rGO. Thus, it can be inferred from the IR and TGA results that GO was reduced by vacuum annealing, but the reduction was incomplete. In contrast, only a slight change can be seen in the Raman spectra (Figure 4d, orange line). As the intensity ratio of the D-band to the G-band ($I_D/I_G$) is a common indicator of defects in carbon materials, we calculated the $I_D/I_G$ for GO before and after annealing. The $I_D/I_G$ for GO and annealed GO were 1.7 and 1.9, respectively. Considering the accuracy of the estimation (see "Peak-fitting analysis of Raman spectra" in the Supplementary Materials for details), we concluded that $I_D/I_G$ slightly increased or did not change as a result of vacuum annealing. As GO contains a large number of defects, the slight increase in $I_D/I_G$ indicates the extension of crystalline domains [38,39]. The IR and Raman spectroscopy results suggest that GO was reduced by vacuum annealing and that rGO still has a highly defective structure (hereafter, the annealed GO-coated MSS is described as rGO-coated MSS). Gas sensing measurements results after vacuum annealing are shown in Figure 5b,c. The sensitivities to the vapors were drastically decreased by vacuum annealing: the peak heights for water, acetone, ethanol, heptane, and ethyl acetate decreased to 3, 8, 12, and 9%, respectively.

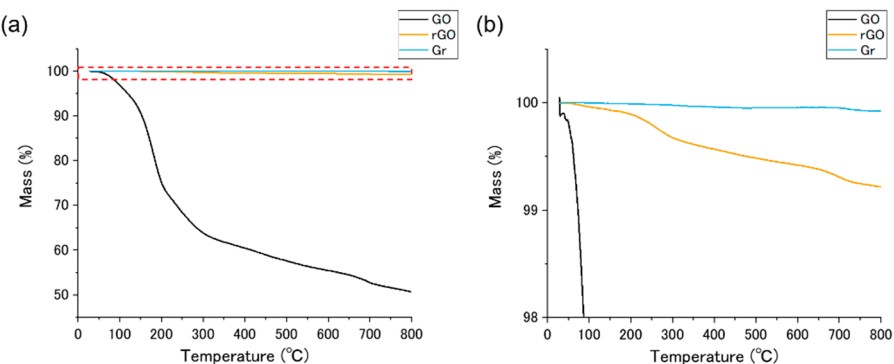

**Figure 6.** Results of thermogravimetric analysis (TGA). (**a**) TGA curves of GO, rGO, and graphite powder (Gr). (**b**) Enlarged figure of the TGA curves indicated by the dashed rectangle in (**a**).

The results of the GO- and rGO-coated MSS suggest that oxygen-containing functional groups play an important role in the properties of graphene-based gas sensing materials. However, rGO still contains a certain amount of oxygen-containing functional groups as thermal annealing at 200 °C cannot fully reduce GO [40]. To investigate the effects of such functional groups, we measured the sensing properties of an MSS coated with graphite powder, which contains few oxygen-containing functional groups. Figure 7a shows an optical microscope image of graphite powder on a Si substrate. The size of the graphite flakes is within the range of 1 to 10 μm. Observation with SEM also confirmed the size of the flakes and multilayered structure of graphite (Figure 7b). Figure 7a also indicates that most of the graphite flakes were not single-layered graphene as they were not transparent. An optical microscope image of MSS coated with graphite powder (Gr-coated MSS) is shown in Figure 7c. The amount of the coating was the same as that of the GO coating. Figure 7d shows the IR spectrum of the graphite powder. Few characteristic features related to functional groups can be seen in Figure 7d. Considering

that the weight loss of the graphite powder was very small as shown in Figure 6b, the graphite powder contains almost no oxygen-containing functional groups. Figure 7e shows a Raman spectrum of the graphite flakes. Unlike the Raman spectra of GO and rGO (Figure 4d), very sharp G- and D-bands can be seen for the graphite powder, indicating that it contains a much lower density of defects than GO or rGO. The 2D-band also confirms that the graphite flakes are not single-layered graphene, because the 2D-band is wider than the G band [41]. To compare the hydrophilicity of GO, rGO, and graphite powder, water contact angles were measured for Si substrates coated with each material. Figure 8a–c shows the results. GO showed the lowest contact angle (36°) of the three materials, indicating that it is the most hydrophilic. rGO and graphite powder showed higher water contact angles than did GO: 76° and 77°, respectively.

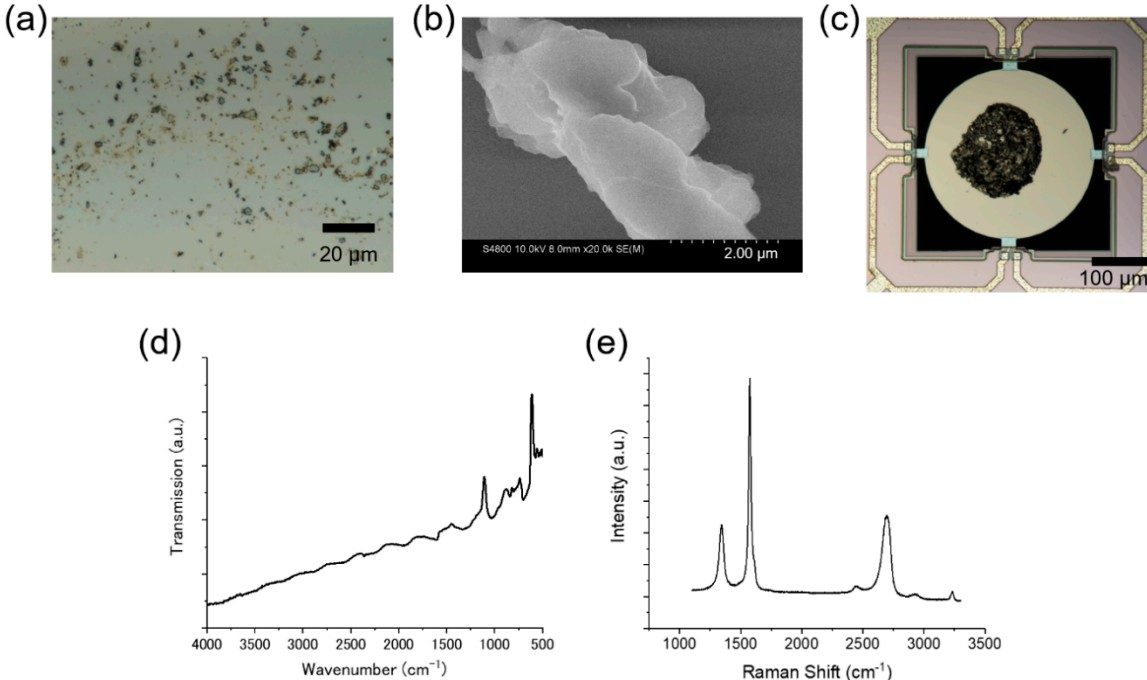

**Figure 7.** Properties of graphite powder. (**a**) Optical microscope image of the graphite flakes on a Si substrate. (**b**) SEM image of the graphite powder. (**c**) Optical microscope image of Gr-coated MSS. (**d**) IR spectrum of the graphite flakes. (**e**) Raman spectrum of the graphite flakes.

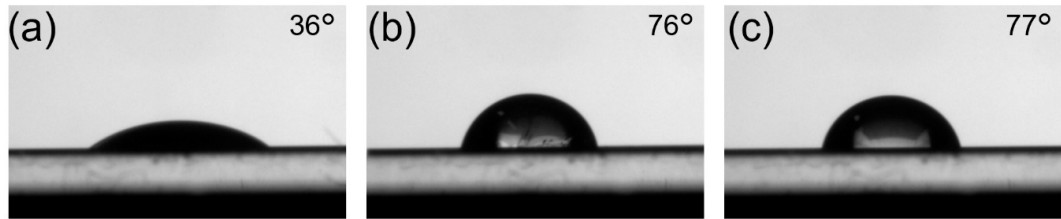

**Figure 8.** Results of water contact angle measurements. Pictures of water contact angle measurements on the (**a**) GO-, (**b**) rGO-, and (**c**) Gr-coated Si substrates.

Gas sensing measurement results for MSS coated with graphite powder (Gr-coated MSS) are shown in Figure 9. Unlike GO-coated MSS, Gr-coated MSS only showed very slight responses to vapors.

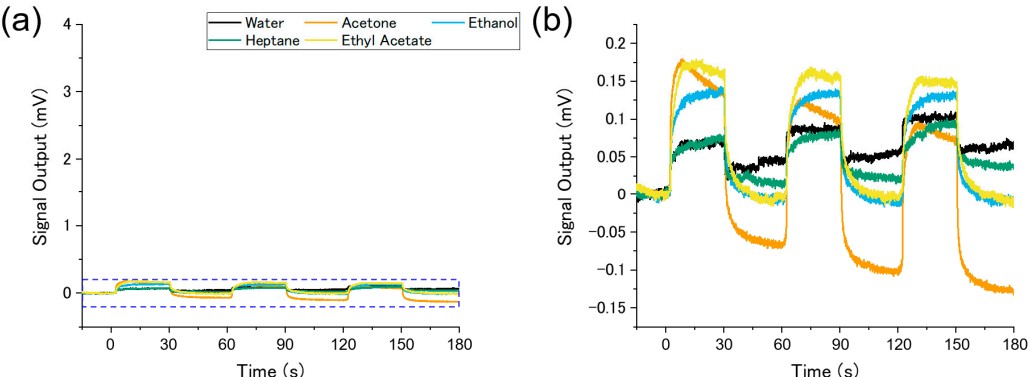

**Figure 9.** Results of gas sensing measurements for Gr-coated MSS. (**a**) Sensing signals of Gr-coated MSS for the solvent vapors. (**b**) Enlarged figure of the sensing signals of Gr-coated MSS indicated by the dashed rectangle in panel (**a**).

### 3.2. Standard Gas Measurements

As GO-coated MSS exhibited the highest sensitivity of the three MSS, we investigated its sensing performance in detecting low-concentration gases in order to assess its practical application. A schematic illustration of the experimental set-up is shown in Figure 2b. In this experiment, the same five gases (i.e., water, acetone, ethanol, heptane, and ethyl acetate) at a concentration of 100 ppm were used as sample gases. Nitrogen gas and a sample gas were alternately injected into a GO-coated MSS with a gas flow rate of 30 SCCM. The time interval of the injection–purge cycle for the standard gas measurements was set at 5 min, which is longer than that for the measurements of solvent vapors. This is due to the intrinsic behavior of MSS, which requires a certain amount of gas molecules to induce a measurable sensing signal [42]. Figure 10 shows the results. Similar to the results obtained with the solvent vapors, the GO-coated MSS exhibited the highest sensitivity to water. However, unlike the gas sensing measurements that were performed with the solvent vapors, the GO-coated MSS exhibited different selectivities to the other four gases. Sensing signals for hydrophilic gases (i.e., acetone and ethanol) were clear, while only small responses to the hydrophobic gases (i.e., heptane and ethyl acetate) were recorded.

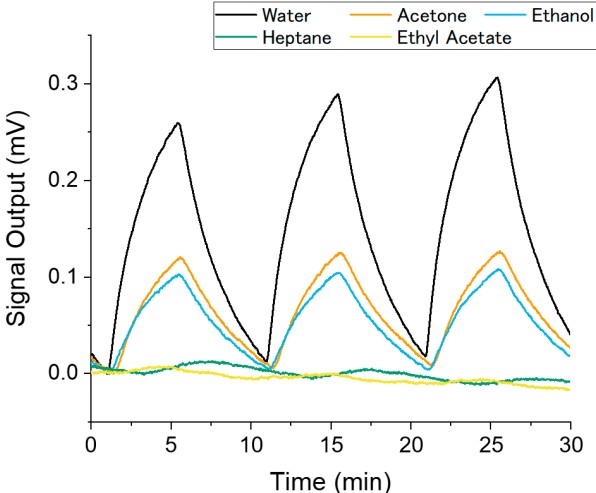

**Figure 10.** Results of standard gas measurements for GO-coated MSS. Sensing signals of GO-coated MSS for the standard gases.

## 4. Discussion

### *4.1. Sensing Mechanism of GO-Coated MSS*

The signal intensities of the MSS for the vapors are summarized in Figure 11a. The signal intensity is defined as the peak height of the final gas injection (the difference between the output voltage at 120 s and that at 150 s). It should be noted that the sensitivity of the MSS is discussed on the basis of the partial vapor pressure—A vapor pressure divided by the saturation vapor pressure. This is because the sensitivity of gas sensors based on gas/solid equilibrium, including MSS, depends on partial vapor pressure rather than absolute concentration [42,43]. In this experiment, the partial vapor pressure of the sample vapors was fixed at 20% by precisely controlling the two MFCs. It is clear from Figure 11a that the GO-coated MSS exhibited the highest sensitivity among the three and is highly selective to water vapor. On the other hand, the rGO- and Gr-coated MSS showed weak sensing signals for the vapors. Considering the structure of each material, these different sensing responses can be ascribed to the existence of hydrophilic functional groups, especially oxygen-containing functional groups. It is well known that GO contains various oxygen-containing functional groups, such as carboxy, epoxy, and hydroxy groups [24]. As such functional groups exhibit high affinities to hydrophilic molecules, especially to water, the GO-coated MSS exhibited high sensitivity to water. Additionally, such functional groups can enhance adsorption of hydrophobic molecules; theoretical studies have reported that graphene functionalized with hydroxy and carboxy groups exhibits stronger binding to methane than does non-functionalized graphene [44,45]. In contrast to GO, rGO and graphite contain few functional groups that can interact with gas molecules, resulting in the low sensitivity of the rGO- and Gr-coated MSS. In addition, water retained in GO may also contribute to the high sensitivity of GO, as Kim et al. reported that such water molecules can enhance absorption of gases [46]. We also measured heptane vapor with a GO-coated MSS before and after annealing the coating at 110 °C for 2 h in vacuum and observed a 40% decrease in signal intensity. These experimental results and associated discussion are described in "Effects of the drying process" in the Supplementary Materials. Because of the retained water as well as the oxygen-containing functional groups, the GO-coated MSS exhibited an even higher sensitivity than did the rGO- and Gr-coated MSS. On the basis of this mechanism, we illustrate the sensing mechanism of GO-coated MSS and rGO-coated MSS in Figure 12.

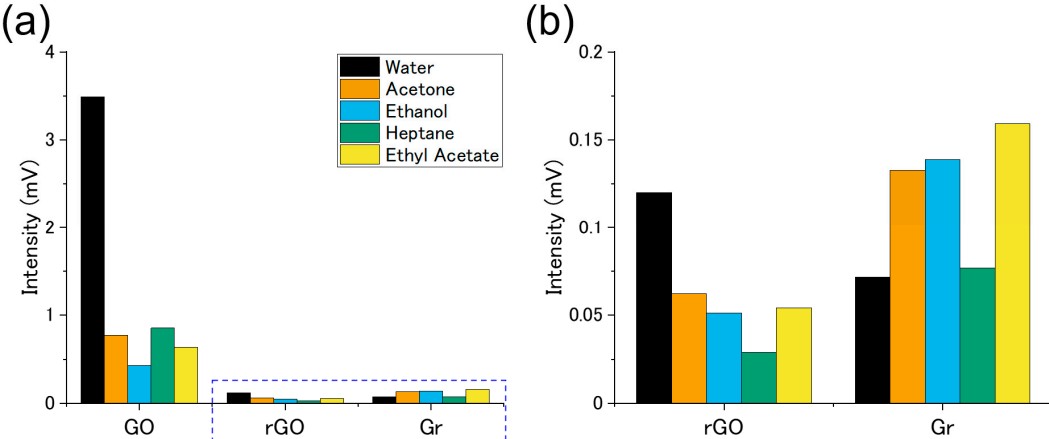

**Figure 11.** Signal Intensities of GO-, rGO-, and Gr-coated MSS. (**a**) Summary of the signal intensities of GO-, rGO-, and Gr-coated MSS for the solvent vapors. (**b**) Enlarged figure of the intensities of rGO- and Gr-coated MSS.

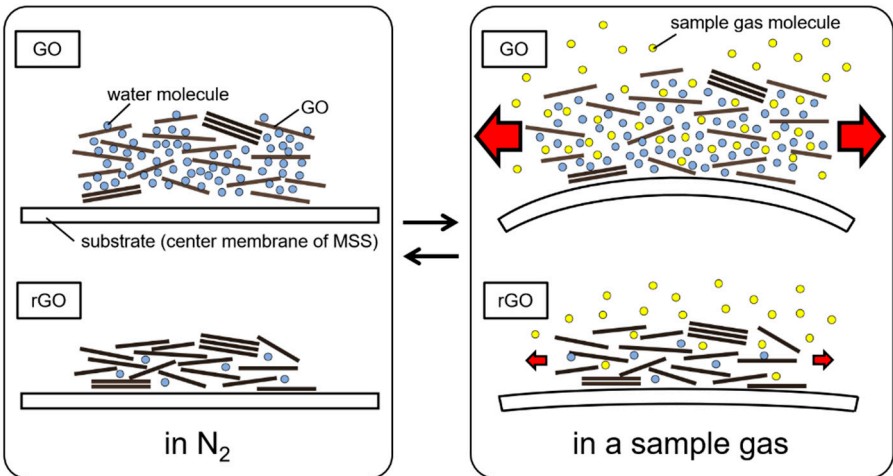

**Figure 12.** Gas sensing model. Schematic illustrations of the sensing mechanism of GO- and rGO-coated MSS.

As MSS detect surface stress caused by the swelling of a sensing material, analysis based on a theoretical model is essential to interpret the sensing properties of a GO-coated MSS. As such a model, a viscoelastic model proposed by Wenzel et al. provides quantitative insight. The viscoelastic model is based on a cantilever coated with a viscoelastic material. In this model, several properties of a gas sensing material contribute to the surface stress on a nanomechanical sensor, especially the partition coefficient of a gas ($K_p$), the relaxed (asymptotic) elastic modulus ($E_R$), and the specific volume of the absorbed gas ($V_a$) [13]. In the steady state, at which the sensing signal levels off, the surface stress ($\sigma$) caused by gas sorption is described as

$$\sigma = -E_R V_a K_p C_g / 3 \tag{2}$$

where $C_g$ is the concentration of a sample gas [12,14]. Although the structure of MSS is different from that of a cantilever-type sensor, this model is adaptable to MSS, because the sensing principle is the same for MSS and cantilever-type sensors. This was confirmed by finite element analysis in previous studies [47–49]. Among these parameters, $V_a$ and $K_p$ reflect the interaction between the gas sensing material and the sample gas ($V_a$ of a gas may change depending on gas sensing materials, because $V_a$ is determined by interactions between a gas molecule and its surroundings). Hydrophilic materials, for example, should exhibit high $K_p$ for water and hydrophilic gases, because the proportion of water molecules in a solid phase becomes larger for hydrophilic materials than for hydrophobic materials. In the case of GO, a high $K_p$ is expected for water and hydrophilic gases because of the hydrophilic nature of GO, and indeed the GO-coated MSS exhibited highest sensitivity to water vapor among the five vapors. Thus, it is suggested that the high sensitivity of GO-coated MSS to water vapor is ascribed to the high $K_p$ of GO for water. This discussion is supported by the result of water contact angle measurements, where GO has a lower contact angle than do rGO and graphite powder, indicating that GO is the most hydrophilic of the three materials; in other words, GO exhibited higher affinity to water than do rGO and graphite powder. However, it seems inconsistent that the sensitivities of GO-coated MSS to other hydrophilic gases (i.e., acetone and ethanol) are not as high as that to water vapor. This relatively low sensitivity of the GO-coated MSS to other hydrophilic gases might be due to the selective permeability of GO to water [50]. Owing to this selective permeability of GO, not only hydrophobic molecules but also hydrophilic molecules are unable to penetrate a μm-thick GO film, even though GO is hydrophilic. Thus, such gas molecules can be absorbed only on the surface of the GO coating, while water molecules can diffuse into the entire GO coating, owing to the low friction of water flowing through the nanocapillaries of graphene laminates [50,51]. Therefore, the total amount of absorbed

molecules in a GO coating should be even higher for water than for other gases, resulting in high sensitivity of GO-coated MSS to water. Although $V_a$ is also an important factor in signal intensity according to Equation (2), the effect of $V_a$ on the high selectivity to water is suppressed, as molecules other than water are unable to penetrate the GO coating. The importance of $V_a$ is discussed in greater detail in "Contribution of $V_a$ in the sensing model" in the Supplementary Materials. As the Young's moduli of GO and rGO are not significantly different [16,17], the contribution of the variation in $E_R$ is presumed to be negligible.

It is also noteworthy that the selectivities of the GO- and rGO-coated MSS were very similar, despite GO-coated MSS being much more sensitive than rGO-coated MSS. Figure 11b focuses on the intensities of the rGO- and Gr-coated MSS. Both the GO- and rGO-coated MSS exhibited a higher sensitivity to water than to the other vapors, whereas the Gr-coated MSS did not show the highest sensitivity to water. As confirmed by the results of the IR and contact angle measurements, GO was not fully reduced by the vacuum annealing; a certain number of oxygen-containing functional groups still remained in rGO [40,52]. Thus, such residual functional groups may have resulted in the high selectivity of the rGO-coated MSS to water. In contrast, the graphite powder should contain an even lower number of such oxygen-containing functional groups than does rGO, resulting in a selectivity very different from that of GO- and rGO-coated MSS.

### 4.2. Detection of Low-Concentration Gases

As sensing responses were observed for measurements of low-concentration gases (Figure 10), the LoD of GO-coated MSS can be discussed on the basis of the signal-to-noise ratio (S/N). The typical LoD is defined as the concentration at which the S/N is 3 [53]. The noise level of the readout system is approximately 5 μV. Assuming that the signal intensity is proportional to the concentration of gases, we can estimate the LoDs for water, ethanol, and acetone at 6, 15, and 15 ppm, respectively. As the sensing signals did not level off in the present measurement period (i.e., five minutes), longer exposure to the gases will improve the S/N and LoD. As a practical application, GO-coated MSS could be used to detect humidity as low as several tens of ppm. Considering that the legal limit for driving is set at several tens of ppm in many countries, GO-coated MSS might be used to monitor breath alcohol content.

### 5. Conclusions

In this study, we investigated the performance of GO as a gas sensing material in nanomechanical sensors operated in the static mode by using MSS as a sensing platform. By comparing the sensing responses of GO, rGO, and graphite powder, we also investigated the sensing mechanism of MSS coated with these carbon-based materials. We have demonstrated that GO is suitable as a gas sensing material in static mode nanomechanical sensors; GO-coated MSS exhibited clear sensing responses to solvent vapors, especially to water vapor. On the basis of the structures of GO, rGO, and graphite, we have concluded that oxygen-containing functional groups enhance the absorption properties of GO, leading to its high sensitivity and selectivity to water. Finally, the sensing responses of GO-coated MSS to low concentrations (100 ppm) of gases were investigated. Clear sensing signals were obtained for water, ethanol, and acetone; the LoDs for the gases are estimated to be 6, 15, and 15 ppm, respectively. Considering the compactness of measurement systems achievable with MSS, GO-coated MSS could help realize portable devices capable of measuring water and hydrophilic gases at a wide range of concentrations. The present study explored the potential of GO as a sensing material in nanomechanical sensors in the static mode and will help expand the possibilities of nanomechanical sensing.

**Supplementary Materials:** The following are available online at http://www.mdpi.com/2227-9040/8/3/82/s1, Figure S1: Results of DLS Analysis, Figure S2: Surface Profile of GO Coating on Si Substrate, Figure S3: Effect of Vacuum Annealing on Surface Profile, Figure S4: Peak Fitting Analysis on Raman Spectra, Figure S5: Properties of Dried GO.

**Author Contributions:** Conceptualization, G.I. and S.O.; methodology, G.I., K.M. (Kosuke Minami), K.S. (Kota Shiba), and S.O.; software, G.I.; validation, K.M. (Kosuke Minami), K.S. (Kota Shiba), G.Y., and S.O.; investigation, G.I. and K.M. (Kosuke Minami); writing—original draft preparation, G.I.; writing—review and editing, K.M. (Kosuke Minami), K.S. (Kota Shiba), K.M. (Kissan Mistry), K.P.M., M.Y., G.Y., K.S. (Koichiro Saiki), and S.O.; supervision, K.P.M, M.Y., K.S. (Koichiro Saiki), and S.O.; project administration, G.Y.; funding acquisition, G.I., G.Y. and K.S. (Koichiro Saiki); All authors have read and agreed to the published version of the manuscript.

**Funding:** This work was supported by the Leading Initiative for Excellent Young Researchers, Ministry of Education, Culture, Sports, Science and Technology (MEXT), Japan; a Grant-in-Aid for Scientific Research (C) [20K05345, MEXT, Japan]; the Iketani Science and Technology Foundation, Japan; the Murata Science Foundation, Japan; a Grant-in-Aid for Scientific Research (A) [18H04168, MEXT, Japan]; the Public/Private R&D Investment Strategic Expansion Program (PRISM), Cabinet Office, Japan; Mitacs-JSPS Summer Program, Japan Society for the Promotion of Science (Japan) and Mitacs (Canada); the World Premier International Research Center Initiative (WPI) on Materials Nanoarchitectonics (MANA), NIMS; and the MSS Forum.

**Acknowledgments:** We thank Keiko Koda, Yuko Kameyama, Eri Sakon (Research Center for Functional Materials, NIMS), and Fusako Hidaka (MANA, NIMS) for the preparation of MSS and the gas sensing measurements. We also thank Makito Nakatsu, Santha Kumara Dissanayake (Research Center for Functional Materials, NIMS), and Takako Sugiyama (CFSN, NIMS) for the materials characterization. We are grateful to Nippon Graphite Industries, Co., Ltd. and Sumitomo Seika Chemicals Company, Ltd. for providing the graphite powder and standard gases, respectively. K. Minami thanks the International Center for Young Scientists (ICYS) program, National Institute for Materials Science (NIMS).

**Conflicts of Interest:** The authors declare no conflict of interest.

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
