# Peer review of "Graphene Oxide as a Sensing Material for Gas Detection Based on Nanomechanical Sensors in the Static Mode"

_chemosensors, doi:10.3390/chemosensors8030082_

Round 1

Reviewer 1 Report

Authors reported the utilization of graphene as a sensing material in static-mode nanomechanical sensors, which was interesting. However, the manuscript still has many flaws. The advices are listed here.

  1. The gas-sensing results show that the GO-based sensor has a selectivity to water. Why not to use Humidity Sensor instead of gas sensor in the title?
  2. The abstract should include main results.
  3. In the part of 2.5, The measurement temperature was kept at 25 °C, while all components including pipes were kept at 37 ° Please elaborate how does it work?
  4. In Figure 2, the SEM image of GO is not clear enough to show the 2D structure of GO. Please characterize again by SEM or TEM.
  5. It will be easy for reader to understand the experimental results if authors explain what the signal output refers to in figure 4.

Author Response

Dear Reviewer 1,

We would gratefully acknowledge the reviewer for his/her critical reading of the manuscript, his/her interests in our work, and his/her essential comments. We believe that we have successfully revised our manuscript, which answers the reviewer’s questions. Please see the attached file, which includes the reply to each comment.

Reviewer 2 Report

This paper described the use of a coated membrane supported by 4 piezoelectric devices as a sensor. The absorption of various vapours into the graphene/graphite coating results in stress on the membrane which can be detected

The introduction and abstract need a brief explanation of what type of MSS is being used. The experimental method should include more detail such as what material the membrane is made of and the authors should consider including a diagram of how it is connected such that the reader can understand what is actually being measured.

The introduction and abstract should also include details of the gases sensed and an explanation that the solvent vapor tests are used to deliver very high concentrations of analyte and the standard gas tests are used to deliver known low concentrations of the same vapors.

The authors refer to operating the sensor in static mode, but it is not until very late on in the manuscript that a description of static mode is given, this description is needed much earlier. Furthermore in the introduction it is stated that the most commonly used nanomechanical sensors are cantilevers - at this point the reader needs to know that this paper does not use simple cantilever devices.

Methods need details of the coating method used for the Si substrate - same as for the MSS?

The description of how the rGO was prepared is needed in the methods section before materials characterisation.

Referring to the analytes as solvents is misleading, they are all commonly used as solvents, but that is not the case in this work they are being used as analytes.

The early data sets were discarded  - this was described as being due to residual gases from previous experiments. This does not inspire confidence in the experiments being well controlled. This could have been eliminated by flushing the system for longer between experiments. Did the authors consider the possibility that the initial data differs from the latter data because the first process of absorption and desorption of the analytes causes morphological changes and therefore alters the sensitivity of the sensor system? Latter cycles then undergo a more repeatable absorption and desorption process.   

Some more discussion of why the low concentration experiments took so much longer to reach equilibrium is needed

Remove the first sentence in the results section

the claim that peak at 5000 nm is small in the DLS data cannot be made as there is no data available above 5000 nm to prove that this is not just the side of a very large peak. Remove this statement.

Describing the coating as a roughly 0.1 um film is misleading. The thickness varies from 0 to 500 nm across the sample. It is not really a uniform film but rather an irregular blob.

There is no discussion of how the incomplete coverage of the MSS membrane affects the sensor. It would be interesting to know how better coverage would affect the responses.

The paragraph after figure 4 is a repeat of an earlier paragraph - please remove.

Why was grahite powder considered - it does not benefit from the large surface area that was used to justify investigating graphene. 

Why is the graphite given a separate section but the GO and rGO are described together? Better to describe all 3 together.

The authors should have made more significance of the contact angle measurements which seem to provide a good test of what is likely to make a sensitive sensor material.

The data in figures 8 and 4b are too small to see. Include data on the same scale as 4a to show the difference in response but also magnify the data in 4b & 8 so it can been observed.

The section on MSS theory line 389-405 should be moved to the introduction.

line 412 clarify that only h2o is adsorbed into the bulk of the material and the other analytes are adsorbed onto the surface only. The discussion refers to hydrophobic and hydrophilic, but water is not considered in this pert of the discussion - this is confusing.

The claim that the humidity sensor can detect 10ppm in a few minute is misleading because the 100ppm experiments ran for 5 minutes each but had not reached equilibrium. 

Author Response

Dear Reviewer 2,

We would gratefully acknowledge the reviewer for his/her critical reading of the manuscript, his/her interests in our work, and his/her essential comments. We believe that we have successfully revised our manuscript, which answers the reviewer’s questions. Please see the attached file, which includes the reply to each comment.

Reviewer 3 Report

In this manuscript, the authors have reported the gas sensing characteristics of GO-based nanomechanical sensors. However, although of the potentially interesting results, this correlation was not well-established and the authors should elaborate more on several points, as suggested below:

Line 185: “Although a few large flakes are observed, most are too small to be seen with the optical microscope.” This is an unnecessary phrase and should be removed.

 Line 187 “Although a peak can be seen at approximately 5,000 nm in addition 189 to the peak near 800 nm, the contribution of the component at 5,000 nm is small.” The authors did not provide any explanation about the second peak. Where does it come from? What it means? The authors should argue about it on the revised manuscript. In addition, Figure 2c should be edited to the size range of interest (100-10000 nm).

The text between lines 211-252 are exactly the same that the text between 253 and 298. This should be revised.

The Figure 2a and 2b should not be shown in the same scale because it makes difficult to the readers to estimate the decrease in sensor response. Besides, the authors wrote "The sensitivities to the vapors were drastically decreased by vacuum annealing." This discussion is too vague and should be improved estimating quantitatively this reduction. In addition, the x axis is elapsed time, so, in all figures it may be started from 0 s.    

Which were the vapor concentrations of tested VOCs and the relative humidity during water test? It is acceptable to discuss the sensitivity using different concentrations for different gases. On the other hand, selectivity results only make sense when compared at the same concentration. Besides, the as called, by the authors, “normalized intensities” in Figure 10b induce the readers to believe that GO response is as high as Gr response for water and ethyl acetate, respectively, which is definitely wrong. GO has an enhanced sensitivity and, maybe, selectivity (depending on the concentration as discussed above) to water while rGo and Gr have low sensitivity and selectivity to all tested vapors. The discussion should be improved taking that in account.

Figure 8, 9 and S4 showed negative values on gas sensing response. What that means?

The authors presented the results and discussed the hydrophilic characteristics of the samples, which is related to the interaction of the liquid water with the surface of the samples. However, it is well-known that the gas sensing response is consequence of the interaction of the vapor molecules with the sample surface, either for water and VOCs.  The authors tried to explain the gas sensing behavior of the material based on its hydrophilic characteristics. But this correlation is not well-established and should be better elaborate. 

The authors have made all the gas sensing measurements using N2 as baseline instead air. Why? Which is the influence of oxygen presence in the gas sensing response? If potential practical applications are aimed, air baseline measurements are required.

In summary, the organization of the manuscript and quality of the writing should be improved as well as new data and better discussions should be included before the manuscript be considered for publication in CHEMOSENSORS.   

Author Response

Dear Reviewer 3,

We would gratefully acknowledge the reviewer for his/her critical reading of the manuscript, his/her interests in our work, and his/her essential comments. We believe that we have successfully revised our manuscript, which answers the reviewer’s questions. Please see the attached file, which includes the reply to each comment.

Round 2

Reviewer 1 Report

Accept.

Reviewer 2 Report

The point raised have all been addressed to a reasonable standard

Reviewer 3 Report

The authors have made significant improvements and the revised manuscript is adequate for publication in Chemosensors.